# Phytoremediation of Heavy Metals: An Indispensable Contrivance in Green Remediation Technology

**DOI:** 10.3390/plants11091255

**Published:** 2022-05-06

**Authors:** Shahnawaz Hassan, Sartaj Ahmad Bhat, Vineet Kumar, Bashir Ahmad Ganai, Fuad Ameen

**Affiliations:** 1Department of Environmental Science, University of Kashmir, Srinagar 190006, India; bhatsabreen32@gmail.com (S.); shahnawazhassan89@gmail.com (S.H.); 2River Basin Research Center, Gifu University, 1-1 Yanagido, Gifu 501-1193, Japan; 3Department of Botany, Guru Ghasidas Vishwavidyalaya (A Central University), Chhattisgarh, Bilaspur 495009, India; drvineet.micro@gmail.com; 4Centre of Research for Development, University of Kashmir, Srinagar 190006, India; 5Department of Botany and Microbiology, College of Science, King Saud University, Riyadh 11451, Saudi Arabia; fuadameen@ksu.edu.sa

**Keywords:** phytoremediation, heavy metals, phytochelatins, pollution, macrophytes

## Abstract

Environmental contamination is triggered by various anthropogenic activities, such as using pesticides, toxic chemicals, industrial effluents, and metals. Pollution not only affects both lotic and lentic environments but also terrestrial habitats, substantially endangering plants, animals, and human wellbeing. The traditional techniques used to eradicate the pollutants from soil and water are considered expensive, environmentally harmful and, typically, inefficacious. Thus, to abate the detrimental consequences of heavy metals, phytoremediation is one of the sustainable options for pollution remediation. The process involved is simple, effective, and economically efficient with large-scale extensive applicability. This green technology and its byproducts have several other essential utilities. Phytoremediation, in principle, utilizes solar energy and has an extraordinary perspective for abating and assembling heavy metals. The technique of phytoremediation has developed in contemporary times as an efficient method and its success depends on plant species selection. Here in this synthesis, we are presenting a scoping review of phytoremediation, its basic principles, techniques, and potential anticipated prospects. Furthermore, a detailed overview pertaining to biochemical aspects, progression of genetic engineering, and the exertion of macrophytes in phytoremediation has been provided. Such a promising technique is economically effective as well as eco-friendly, decontaminating and remediating the pollutants from the biosphere.

## 1. Introduction

Environmental contamination has become a grave public health problem impacting human sustainment and survival across the globe [1]. Pollutants degrade environmental quality, the majority of it being contributed by toxiferous metals. The acute danger accompanying toxic metals on human wellbeing has been recognized for an extended period; still, their exposure to humans lingers and is aggregating in numerous areas of the universal domain. Heavy metal (HM) exposure can severely impact human health and can sometimes prove fatal [2]. Global industrial processes are believed to be the reason for global HM pollution [3,4]. Heavy metals (HMs) can easily become amassed in the environment. For example, when the amount of HMs increases above the standardized limits, it results in bio-magnification via the food chain, affecting all the biota of the planet. The removal of these metal pollutants, thus, becomes significantly important to reduce the threat to all forms of life as well as to our natural surroundings. Many processes/techniques, such as reverse osmosis [5], chemical precipitation [6], ion exchange [7], adsorption, and solvent extraction [8], have been put into place to eliminate the HMs from the environs. However, these techniques involve significant maintenance functionalities and expenses and are generally not sustainable. Phytoremediation offers one of the environmentally suitable approaches to overcome toxic metal pollution (Figure 1) as a cheap and alternative way to decontaminate the HM-contaminated sites [9]. The technique of phytoremediation is widely accepted worldwide owing to its lower cost in comparison to traditional remediation methods [10,11]. Such a technique has minimal impact on the environment because no change in the soil structure is required [12]. The area can be utilized again for agricultural activities or as farmland after phytoremediation is complete [13]. This promising technology uses hyperaccumulators to eradicate metal toxicity from the contaminated sites [14]. The removal capacity of metal ions by plants is also influenced by an important parameter known as the bioconcentration factor (BCF). It offers an index of the proficiency of the plant to amass the metal with respect to the metal concentration in substrate. The BCF varies with the type of medium and selection of plant species. Hyperaccumulators tend to grow roots in areas of high metal concentrations, having high levels of uptake into root cell symplasm and reduced root vacuolar transport [15]. Hyperaccumulators have a suite of characteristics, such as a BCF greater than one, shoot–root metal concentration quotient greater than one, and phenomenal metal tolerance, greatly due to effective detoxification [13,16]. Some of the hyperaccumulators have been studied for their high accumulating HM potential (Table 1). An attempt has been made to provide a detailed review regarding the various aspects of phytoremediation. An insight into the exertion of different macrophytes that can be utilized for the removal of pollutants, particularly HMs from the environment, has also been elaborated in detail. 

## 2. Heavy Metals in the Environment

HMs in environs are significantly contributed to by both natural (geological activities) as well anthropogenic activities. The central basis of HM pollution is the haphazard and continuous release of metal-rich industrial wastes [83]. The expulsion from metal-based industries, especially leather industries, is a grave environmental concern, especially for soil and water; thereby, an immediate well-defined approach for its abatement is of paramount importance [84]. Similarly, the unnecessary consumption of pesticides and fertilizers on agricultural soil for maximum output has tremendously amplified the standard limits of HMs in soil, mostly due to the ever-swelling world population [85]. This has raised significant apprehensions about their possible implications for the environment [86]. The other known basis of HM pollution is the application of wastewater as an irrigation source and transportation that has led to the accretion of abundant HMs in the subsurface of the soil. Activities such as road maintenance and deicing operations produce groundwater and surface pollutants, hampering environmental wellbeing [87].

## 3. Process of Phytoremediation

The technique of phytoremediation is the blend of two words “phyto” which means “plant” and the Latin suffix “remedium” which means to “restore”. The process of phytoremediation uses both natural as well as transgenic plants to remediate the polluted ecosystems [88]. Over the years, the process of phytoremediation has gained tremendous significance in terms of scientific and commercial considerations [89]. The exertion of hyperaccumulators for degradation, extraction, absorption of toxic metals and other harmful pollutants was first presented in 1983 [90]. The process employs diverse collections of phytotechnologies that use both natural as well as genetically modified plant species for eliminating the environmental effluence [90,91].

The phytoremediation process can be achieved by using both in situ as well as ex- situ techniques. The in situ application technique is more frequently used as it decreases the proliferation of pollutants in soil, water, and airborne waste, which automatically diminishes the risk to the neighboring environment [92]. The in situ technique has another major advantage in that multitudinous pollutants are treated on a particular site without the requirement for a disposal site. The in situ technique also decreases the range of pollution by checking different soil parameters, such as erosion and leaching. Similarly, the ex situ method of bioremediation involves the removal of contaminated soil and subsequently transporting it to another site for treatment. Factors such as the graphical location of the contaminated site, cost of treatment, pollutant type, and severity of pollution are the main criteria for ex situ bioremediation technique. Ex situ bioremediation techniques are easier to control and are used to treat a wider range of toxins and soils. However, the ex situ techniques of phytoremediation appear to be more expensive in comparison to in situ techniques. Both these mechanisms of phytoremediation show significant differences in their experimental controls and the consistency of the process outcome. Post-treatment, phytoremediation proves to be economically efficient in comparison to other remediation techniques [93], as it is a simple, non-laborious technique requiring no installation of special equipment. The process can be employed to an enormous extent where other commonly employed techniques prove inefficient and extremely expensive [94]. The applicability of hyperaccumulator plants has been analyzed recently and this invigorated more research concerning the molecular basis of phytoremediation [95].

For the implementation of the phytoremediation technique for the HM remediation, two defense strategies that can be adopted are avoidance and tolerance [96]. Plants utilize these two approaches to balance the concentration of HMs beneath their lethal threshold levels [97].

Avoidance is a process where plants use root cells to limit and restrict the uptake and movement of HMs into the plant tissues [98]. Such a process involves various defense mechanisms (root sorption, metal precipitation, and exclusion) [98]. When plants are exposed to HMs, the root sorption process is involved in their immobilization. A wide range of root exudates acts as a HM ligand to form HM complexes in the rhizosphere, through which the bioavailability and lethality of HMs is restricted [98]. Similarly, the exclusion barriers that occur between the root and shoot system also restrict the accessibility of HMs from the soil to the roots. Moreover, arbuscular mycorrhizas can also act as exclusion barriers for HM uptake through the absorption, adsorption, or chelation of HMs in the rhizosphere [97]. HM embedding in the plant cell wall is an additional avoidance appliance, as the pectin groups (carboxylic groups) in the cell wall act as cation exchangers to limit the entry of HMs in the cells [99].

The tolerance strategy is implemented by the plants once a HM ion intrudes into the cytosol to cope with its toxicity, accomplished by the processes of inactivation, metal chelation, and HM compartmentalization [98]. Through chelation, the concentration of HMs is reduced by various organic and inorganic ligands in the cytoplasm [100]. After chelation, the HM ligand complexes are transferred from the cytosol into inactive compartments (vacuole, leaf petioles, leaf sheaths, and trichomes) where these are stored without toxicity [101].

If there is a high accumulation of HMs, the above strategies are sometimes not adequate to remediate the contaminated sites as HM accumulation can trigger the generation of reactive oxygen species (ROS) in the cytoplasm causing oxidative stress [102]. To cope with such a situation, antioxidant enzyme superoxide dismutase (SOD), catalase (CAT), peroxidase (POD), and glutathione peroxidase (GR) as well as non-enzymatic antioxidant compounds (i.e., glutathione, flavonoids, carotenoids, ascorbate, and tocopherols) are utilized by the plant cells to trigger ROS scavenging [102,103]. Hence, the antioxidative defense mechanism is highly crucial and imperative concerning HM stress.

## 4. Phytoremediation Approaches

Phytoremediation follows various contrivances such as phytoextraction, rhizofiltration, rhizodegradation, phytostabilization, phytodegradation, and phytovolatilization (Figure 2) during the interaction and accumulation followed by the intake and accrual of HMs in the plant [90]. The mechanisms involved are concisely defined and elaborated below.

### 4.1. Phytoextraction

Phytoextraction encompasses the intake of HMs and their movement to higher parts of the plants, such as shoots, leaves, stems, and other parts [104]. A survey of the literature shows that numerous hyperaccumulator metallophytes have significant potential that can be utilized for the treatment of HM-contaminated soils [105]. Hyperaccumulator metallophytes can amass HMs in their higher parts in concentrations between 100 and 500 times more than other plants without affecting their development and functioning [106]. However, the mechanism of heavy metal accumulation by the hyperaccumulator metallophytes is still understudied and, thus, can be studied and further elaborated to understand the fundamental process of heavy metal accumulation [107]. The efficiency of phytoextraction is regulated by the parameters, such as the BCF and translocation factor (TF); hence, successful phytoextraction is acclimatized by improving these factors in combination with increasing the import into epidermal or cortical cells, or export from pericycle or xylem parenchyma cells into the stellar apoplast, and converts the metals into the less harmful state [108]. The nature and quantity of chelators determine the rate of HM absorption by vacuole sequestration by hyperaccumulators [104]. Artificial chelates are now being added to enhance mobility and uptake, thereby improving the efficiency of phytoextraction.

Two key characteristics that define the phytoextraction perspective of plant species is their capacity to accumulate HMs and above-ground biomass; therefore, plants that hyper accumulate HMs in above-ground parts and plants with high above-ground biomass production are employed for phytoextraction [78,109]. For successful phytoremediation of HMs, finding effective hyperaccumulators holds the key, and more than 450 plant species have currently been identified as potential metal hyperaccumulators [110]. It has also been revealed that some of these species have the potential to accumulate more than two elements, for example. *Sedum affredii* [111]. Currently, scientific investigations are underway around the world to expand the effectiveness of phytoextraction where novel hyperaccumulators are targeted to improve understanding of their biological conduits. There are some plant families, such as Asteraceae, Brasicaceae, Euphorbiaceae, Fabaceae, Flacourticeae, and Violaceae, that have been proven to accrue greater concentrations of HMs [112]. Among these, species belonging to the Brassicaceae family have shown enormous potential to remediate and scavenge HMs, such as lead (Pb), cadmium (Cd), zinc (Zn), and nickel (Ni) [109]. Different *Brassica* species have been investigated for HM accumulation by researchers across the world. These include *Brassica juncea* L., *Brassica oleracea* L., *Brassica compestris* L., *Brassica juncea* L., and *Brassica napus* L. [112]. Among these, *Brassica juncea* L. has shown tremendous potential to remediate HMs, such as Cd, Cr, Cs, Cu, Ni, Pb, U, and Zn [113]. Similarly, another study carried out at Florida University on plant species *Pteris vittata* (Chinese brake fern) has indicated that it can be a potential candidate for arsenic (As) removal (3280–4980 ppm) [114,115]. To remediate the radionuclide-based soil, sunflower (*Helianthus annus*) has emerged as a feasible hyperaccumulator plant to remediate soil contaminated with cesium-137, strontium-90, and uranium [116]. One of the advantages of phytoextraction is that it can be used as an energy source when used in combination with a biomass, such as bio-ore, and can form the base for phytomining [117]. Furthermore, when the mechanism of phytoextraction, which involves the processes of absorption and transport capacity of the hyperaccumulator, is understood fully, mathematical modeling of HM bioaccumulation can be advanced [118]. As well as metallophyte plants, metallophyte algae (Table 2) can also be put to use for heavy metal removal. Algae is involved in the absorption process by taking the heavy metals by adsorption and into the cytoplasm by chemisorption [119].

However, there are certain concerns to consider, such as the usage of edible crops for phytoextraction. Such exercise should be avoided as HMs bioaccumulate in the plant’s edible part, thereby intruding into the food chain, which can have deleterious impacts on human health. Hence, it is imperative to select non-edible hyperaccumulators for the efficient and safe phytoremediation of HMs.

The biomass containing higher heavy metal concentration collected after the phytoextraction process may present a hazard to human well-being and the environment. There are a few approaches, such as neutralization techniques, that aid in storing the polluted biomass material in landfills [13]. Pyrolysis of contaminated biomass in waste processing installations can be another neutralizing approach [13].

### 4.2. Rhizofiltration

Rhizofiltration utilizes roots to absorb, retain, and settle metal contaminants within the roots, ensuring limited movement of these contaminants into different environments [136]. In the root microbiome, the environmental factors, such as pH in the rhizosphere, root exudates, and root turnover, play a vital role in the settling of metal contaminants on the root surface. As soon as the plant has taken up all the metal pollutants, the plant can be easily collected and disposed of in a safe site [137]. In this process, the plant and microbial community have a symbiotic association. The plants increase the microbial activity while microorganisms decontaminate the metal component. Bacteria generally used in rhizoremediation are *Pseudomonas aeruginosa*, *Mycobacterium* spp., and *Rhodococcus* spp. [138]. Usually, wild-types of microorganisms are selected for this process, which does not entail the use of transgenic bacteria. Rhizoremediation simply involves remediation that revolves around roots, microbes, and rhizospheric soil. However, the plants employed in the rhizofiltration technique must have the potential to yield a wide-ranging root system, must accumulate HMs in greater concentrations, should be easy to handle and harvest, and have a truncated preservation budget [91]. Plants produce a niche for rhizosphere microorganisms to accomplish HM transformation. Soil contaminated with organic compounds is degraded by this method. Environmental variables such as pH, temperature, soil, and plant species have a very important role in rhizoremediation success [26].

For rhizofiltration, both aquatic as well as terrestrial plants can be employed. Aquatic species (hyacinth, *Azolla,* duckweed, cattail, and poplar) are frequently utilized for the remediation of wetland water mostly because of their high accumulating capacity, high tolerance, and greater biomass production [139]. Similarly, terrestrial plants (*B. juncea* and *H. annus*), owing to their larger hairy root system, exhibit high capability to cumulate HMs during rhizofiltration [140]; investigations have demonstrated that sunflower has tremendous ability to decontaminate Pb-contaminated sites. Similarly, Indian mustard is believed to eliminate greater concentrations of Pb (4–500 mg/L) [92].

Scientific investigations are proceeding at a progressive rate to ameliorate the proficiency of rhizofiltration technology. Different experimental setups have reported that young seedlings show greater capacities to remove HMs from water [141]; a technique commonly called blastofiltration. Through data depiction, it has been revealed that for few metals, such a technique can out-compete the rhizofiltration; however, the greatest benefit associated with rhizofiltration is that it can be applied both in situ as well as ex situ. For aquatic systems with high heavy metal pollution load, the rhizofiltration process is not considered feasible, and it also has drawbacks such as drying, composting, and incineration.

### 4.3. Rhizodegradation

Rhizodegradation involves the biodegradation of the organic pollutants in the soil accompanied by rhizospheric microbes that secrete specific enzymes that degrade or transform exceedingly contaminated organic pollutants into less detrimental forms. The process of rhizodegradation is enhanced as these organisms draw out the essential constituents (nutrients) from the root secretions of the plant, that upsurge the plant efficacy and accelerate the extraction and amputation of pollutants by the plant [142]. One of the important features of rhizodegradation comprises the dissolution of the pollutant at its site; it focuses on the complete mineralization of the organic pollutant following compound translocation to the plant or atmosphere [143]. The process of rhizodegradation has some drawbacks, which include the fact that it is a time-consuming process occurring at a slow pace and is effective only up to a certain depth, usually from 20–25 cm. Rhizodegradation is influenced by soil type and selected plant species [144].

### 4.4. Phytostabilization

The process of phytostabilization or phytorestoration decreases the contaminant movement, thus, inhibiting their passage into underground water, and prevents bio-magnification [145]. The process mainly relies on the utilization of specific plants for the steadiness of contaminants in polluted environments [27]. In contemporary times, HM stabilization by adsorption, binding, or co-precipitation with soil additives (biosolids, manures, and composts) has been extensively investigated in the last decade [146]. Such a remediation exertion has proven successful in decreasing the movement of pollutants in soil environments [147]. It stabilizes contaminants and prevents the contaminants polluting streams, lakes, and ponds and, thus, prevents wind and water erosion. It not only enhances the hydraulic capability for the vertical movement of pollutants but also lessens the pollutant mobility by physical and chemical root absorption. 

The process results in the formation of insoluble compounds in the rhizosphere [148]. The metallophytes are used, successfully reclaiming the sites contaminated with pollutants, and are suitable for the removal of metals, such as Pb, As, Cd, Cr, Cu, and Zn [149], and are very convenient for the areas that are severely contaminated and had occupied large spaces [150]. Phytostabilization is only a management tactic for the inactivation and immobilization of the potentially deleterious contaminants. It only restricts the movement of the metal ions, and it is not an enduring management as contaminants continue to persist in the soil [151]. For phytostabilization to operate successfully, the plant should grow rapidly with a large life span and must be able to adjust to the soil conditions [152]. Many studies have shown that medicinal and aromatic plants can be employed for the elimination of Pb, Zn, Cd [153,154,155]. Alimurgic species (*Cichorium intybus* L. and *Taraxacum officinale*) can be utilized as phytostabilisers for zinc and cadmium removal, respectively [156].

Phytostabilization has a notable advantage of being a technology with easy execution and operating costs.

### 4.5. Phytodegradation

In phytodegradation, organic pollutants are broken down after being sequestered by the plant through various metabolic processes, or degraded by the enzymes involved in the metabolism of the plant [157]. The enzymes involved in the pollutant breakdown are dehalogenase, peroxidase, nitroreductase, nitrilase, and phosphatase [158]. It involves the direct uptake of contaminants into the plant tissue through the root system and primarily depends on uptake efficiency, transpiration rate, and other physical and chemical properties of the soil. Sites affected by organic contaminants, such as herbicides and chlorinated solvents, can be decontaminated by phytodegradation [159]. It can also be employed for the recovery of both surface and ground waters [93]. Different plants can be utilized in this process; sunflower (*Helianthus annus*) for methyl benzotriazole [160] and *Leucocephala* for ethylene dibromide [161] have been widely used.

There are some limitations of this process as the soil must be three feet deep while groundwater should be within ten feet of the surface. Chelating agents are needed to augment the plant uptake by binding the soil particles with the contaminants [162].

### 4.6. Phytovolatilization

Phytovolatilization is a transformation of pollutants into different volatile compounds into the atmosphere via transpiration with the assistance of the stomata [94]. Plants such as *Nicotiana tabacum*, *Crinum americanum*, *Triticum aestivum*, *Arabidopsis thaliana*, *Bacopa monnieri*, and *Trifolium repens* are commonly used plants for phytovolatilization [163]. It can be achieved directly or indirectly. Direct volatilization involves the volatilization of volatile organic compounds (VOCs) by the stem and leaves while indirect volatilization occurs due to plant root interactions in the soil [164]. Phytovolatilization degrades organic contaminants, such as acetone, phenol, and chlorinated benzene (BTEX) [165]. Mercury (Hg) and selenium (Se) show the most encouraging results in the phytovolatilization process [166]. Although it is a slow process, the addition of novel plant species with extraordinary transpiration rates and enzymes such as cystathionine-V-synthase can be employed to enhance the remediation of S/Se volatilization [167,168]. Poplar trees volatilize 90% of trichloroethylene (TCE) after uptake from soil [169]. Transgenic yellow poplar (*Liriodendron tulipifera*) has also been used to remediate Hg. It has been successfully employed to remediate Hg with results showing a 10-fold increase in removal efficiency as compared to non-transgenic plantlets [170]. Currently, with the help of phytovolatilization, radioactive isotopes of hydrogen (tritium) are decayed to stable helium [171,172]. Moreover, microorganisms facilitate the dilapidation of organic compounds in the rhizosphere [173]. The greatest benefit of phytovolatilization is that it hardly requires extra management once the plantation is completed. Moreover, it maintains the soil texture and causes the least disturbance to the soil [93]. Among all the techniques of phytoremediation, phytovolatilization is very contentious [174].

Phytovolatilization as a remediation approach does not decontaminate the environment completely; it only facilitates the pollutant transfer, which can sometimes contaminate the ambient atmosphere as they rise from the soil. Furthermore, these can be redeposited back into the soil with precipitation [175]. This demands a serious assessment of potential risks that could be associated with its applicability in the field.

### 4.7. Phytodesalination

Phytodesalination, a recently engineered and emerging technique, employs halophytic plants to remediate the saline soils and is the most commonly employed biological method for such decontamination [78]. Compared to the other phytoremediation techniques, very little is found in the literature regarding this process. Halophytes are considered to be naturally well-adapted to HMs in comparison to glycophytic plants [176]. The Phytodesalination capacity of the plant depends on the species as well as on the soil properties, such as salinity, sodicity, and porosity, and other climatic factors, particularly rainfall [177]. While going through the literature survey, it has been reported that two halophytic plants, namely *Suaeda maritime* and *Sesavium partulacastrum* can remove almost 504 kg and 474 kg of NaCl, respectively, from one hectare of saline soil in a four-month period [178]. It has been found that desalination studies of halophytic plants show promising results in the remediation of soil affected by sodium (Na^+^) and chloride (Cl^−^) ions. This bioremediation technique is not suitable for the decontamination of soils polluted with HMs and polycyclic aromatic hydrocarbons (PAHs); however, it is promising for salt-affected soils [179].

Plants that utilize their living biomass to accumulate heavy metals have attracted greater research attention worldwide during recent decades. Although hyperaccumulators have been employed for HM removal, hyperaccumulators of Pb, Cu, Co, Cr, etc. still remain largely unconfirmed and require further scientific exploration. This can be achieved by using standard methods for confirming the reliability of analytical data concerning metal and metalloids [180].

## 5. The Progression of Genetic Engineering

The exertion of genetic engineering has proved a key contrivance for ameliorating the phytoremediation capabilities of plants towards HM pollution. A foreign source of the gene from organisms with the help of genetic modification is shifted and installed into the genome of the target plant followed by DNA recombination that confers particular traits to the plant in a shorter space of time. In such a process, genes of notable interest from hyperaccumulators to plant species that are sexually incompatible species can be transferred, which is otherwise not possible using traditional breeding methods [181]. Exertion has shown a significant promise in the field of phytoremediation. However, the gene selection should rely on the information and acquaintance of the HM tolerance and accretion mechanism of plants. HM tolerance to augment antioxidant activity can be realized by the overexpression of genes tangled in the antioxidant mechanism [182]; encoding metal ion transporters, including zinc iron permease (ZIP); metal transporter proteins (MTP); the multidrug and toxin extrusion protein (MATE); HM ATPases (HMA). Similarly, genetic engineering can be employed to promote the production of metal chelators that will enhance HM uptake and translocation [183].

Though the application of genetic engineering has shown notable prospects in phytoremediation, a few setbacks remain to be addressed. Owing to the complications of decontamination and HM accumulation, the genetic manipulation of several genes to enhance the required traits can be time-consuming and less successful. In some parts of the world, plants that are genetically modified find it difficult to gain permission and approval due to the concerns that are associated with their use, raising concerns for food and ecosystem safety. This demands alternative approaches that could augment and enhance the performance of plant species used in phytoremediation once genetic engineering is impracticable.

## 6. Factors Affecting the Metal Uptake

HM accumulation by the plants is affected by many factors (Figure 3), such as plant species, pH, root zone, cation exchange capacity (CEC), [184], the addition of chelators [185], and temperature [186]. The impact of these environmental variables is described as follows:

**Plant species**: Plant species with different potentials for various remediation processes are chosen. Processes such as rhizodegradation, rhizofiltration, and phytostabilization mainly place emphasis on faster growth in terms of root depth, plants mass per unit volume, surface area, and lateral extension [187]. For example, *Robinia pseudoacacia* can be successfully used in an ecological manner to remediate sterile dumps because it is able to extract and remove significant quantities of HMs from sterile material [188]. However, the complete phytoremediation of sterile material could be achieved in a couple of years. For the accumulation of contaminants, plants must be able to store more, hence, require bulky root mass [189]. The plant species should be involved in rapid volatilization, transpiration, increased metabolism, and immobilization of various metal contaminants [188]. The rhizobium should facilitate microbial growth by releasing root exudates and enzymes. Furthermore, plants should pose a high level of capability for remediation, adequate storage and transportation, higher growth rate and good biomass yield, high tolerance of waterlogging, and resistance to high pH and salinity [190].

**pH:** It is considered as one of the utmost aspects affecting the solubility and retention of HMs in the soil. At a higher pH, greater retention and decreased solubility occurs [191], whereas low pH increases the accessibility of hydrogen ions [192]. For example, Pb absorption by plants is highly affected by the pH. To reduce the Pb uptake by the plant, soil pH is adjusted with the aid of lime to levels between 6.5 and 7.0 [193]. Plants can enhance their bioavailability using root exudates altering rhizospheric pH and upsurge the solubility of heavy metals [98]. The metal is then sorbed at the metal surface and moves into the root cells through the cellular membrane using apoplastic (passive diffusion) and symplastic (active diffusion) pathways [194].

Soil pH and soil characteristics strongly influence the solubility of metals. Under acid and oxidizing environments, most of the HMs are readily mobile and are strongly retained under alkaline and reducing conditions [195]. HMs, such as Pb, Zn, Cd, Cu, Co, and Hg, are more soluble from pH 4–5 than in the range from 5–7 [196]. However, certain metals, such as, As, Se, and Mo, under acidic conditions are less soluble due to their anionic nature. Soil pH affects metal adsorption and it has been reported that initial metal concentration influences the metal absorption and equilibrium soil pH [197]. Applications of soil amendments to contaminated soils can help in adjusting pH, which will ultimately increase the metal desorption from soil-to-soil solutions. 

Further research is necessary to investigate the factors that influence soil pH changes in the rhizosphere as it significantly reduces the risk of contaminants leaching down into the soil profile. The elucidation of the processes involved will aid in the documentation and possibly the synthesis of new soil and foliar amendments to hasten the phytoremediation process.

**Root Zone:** The root zone plays a substantial part in phytoremediation as it absorbs and metabolizes the contaminant inside the plant tissue or by degrading the contaminant by releasing the enzymes [188]. The root zone is vital in determining the rate of remediation. For example, the fibrous root system has abundant fine roots that cover the entire soil and provides a higher surface area that enhances the maximum contact with the soil [198]. Similarly, the detoxification of soil contaminants by plant enzymes exuded from the roots is another phytoremediation mechanism [199].

**Cation exchange capacity:** CEC measures how many cations can be retained on soil particle surfaces or the rate of adsorption between various metals on the soil interface. As the investigation carried out by the scientific community has indicated, with the addition of Pb and Cu, calcium absorption is reduced [200].

**Addition of Chelators****:** The chelating agents augment or accelerate the uptake of HMs, thus, it is known to be responsible for induced phytoremediation [201]. Chelates have been employed to upsurge the solubility of metals that could considerably increase metal accrual in plants. The addition of chelates, such as ethylene diamine tetraacetic acid (EDTA) to Pb [II]-contaminated soils increases its solubility [185]. The accrual of HM uptake can be influenced by the progressive increase in biodegradable physiochemical properties, such as chelating agents. However, the application of modern synthetic chelating agents has a serious drawback as there is an increased risk of the leaching of contaminants into the soil [202]. The uptake of HMs is affected by the presence of ligands and influences the leaching potential of metals below the root zone [203].

**Temperature:** Soil temperature is a remarkable factor that affects the metal accretion by plants [204]. For instance, at a high temperature and low soil pH, a substantial proliferation of cadmium and zinc contents of the sorrel and maize shoot has been reported [205].

## 7. Plant Assortment Benchmarks for Phytoremediation

Factors such as root complexity, soil pollutants, soil, and regional climate play a key role in phytoremediation. Many investigations have reported that plants with smaller developing periods as compared to perennial plants are a superior selection that can be utilized in phytoremediation [206]. Similarly, it has been suggested to employ plant species that are adjusted to the regional or local soil characteristics of the area in which decontamination is to be carried out [207]. The non-invasive plant species should be selected as they are intrinsically adapted to tolerate stress conditions of the area; these also have low preservation costs. Moreover, the native plants are environmentally and human friendly as compared to the alien species [208]. It has also been stated by various scientific studies that grasses have speedy growth, enormous biomass, durable resistance, and proficiency to decontaminate different sorts of soil in comparison to trees and shrubs [209].

## 8. Biochemcial Aspect of Phytoremediation

With the progress of molecular technologies, the knowledge of the principles behind phytoremediation, such as hyperaccumulation, has vastly improved [210]. The metal accumulation occurs in different parts of a plant (roots, stems, leaves, seeds, and fruits) according to the specificity of each process [211]. HMs, such as Pb, Zn, As Cr, Cd, Hg, etc., when taken by the plant, disrupt the pigments or enzyme processes by producing ROS, which causes oxidative stress and interferences in the electron transport chain. The oxidative stress results in:Lipid peroxidation;Biological macromolecule deterioration;Membrane dismantling;Ion leakage;DNA strand cleavage.

Interestingly, there are different enzymes involved in oxidative stress breakdown, however, among all these, glutathione (GSH) plays a noteworthy role as it directly takes the free radicles [212]. The whole process is catalyzed by ATP-dependent processes and gamma-glutamyl cysteine synthetase (ƴECS) and glutationine synthetase [89]. The SOD displays a vital role by dismutating the oxygen radicle (O_2_)^−^ to an oxygen molecule (O_2_) and hydrogen peroxide (H_2_O_2_). CAT is responsible for the conversion of H_2_O_2_ to water (H_2_O) and oxygen (O_2_). It functions as a protein-compatible hydrotype, ROS Scavenger, osmoprotectant, and regulator of cellular redox status. Due to stress triggered by the heavy metals, mitrogen-activated protein kinase (MAPK) and other stress-responsive genes are activated [213]. The MAPK pathway is used in triggering intracellular targets by using extracellular signals in eukaryotes [214]. Cadmium and copper activate four MAPKs (SIMK, MMK21, MMK3, and SAMK) in *Alfalfa* whereas one kinase (ATMEKKI) is induced by Cd in *Arabidopsis* and it induces (OSMAPK2) in rice. However, it is not evident whether the process of activation occurs directly by heavy metals or through ROS, which is also responsible for MAPK pathway perturbation. The studies for the cadmium and copper transduction pathway indicate that both ROS and calcium accumulation are responsible for triggering the MAPK pathway. MAPK responses vary with the type of plant involved and are also influenced by the nature of metal. Furthermore, the phytohormones also play an imperative role in activating responsiveness to heavy metals. The phytohormone either directly activates genes or they take part in any reaction, or both processes are involved [215]. Metal-binding protein metallothioneins (MTs), phytochelatins (PCs), and organic ligands take part in the binding, immobilization, and conversion of toxic metals into less harmful states in the above and ground parts of the plant [34,90]. Upon exposure to heavy metals, the plants release PCs and MTs for decontamination of the metals [216]. The MTs are believed to primarily chelate nutrient metals for their respective functions to defend plants from the impact of noxious metal ions [217]. For instance, a transgenically produced tobacco plant with 32 amino acids results in modest levels of Cd (II) resistance and accumulation [218]. Previous studies on plants identified PCs as vital chelators which play important role in phytoremediation [219]. PCs act as precursors to antioxidative mechanisms [220]. The assimilation of Cd in *B. napus* increases the generation of PCs [221]. This process was shown by *B. juncia,* which showed the over-expression of bacterial glutathione synthetase (GS) [137]. Increased concentrations of glutathione and phytochelatins have been detected in transgenic *B. juncia* plants and there is more Cd (II) tolerance and accumulation relative to controls. The change in GSH and PCs concentrations has substantial potential for increasing the HM accumulation by plants. Transmembrane transporters such as zinc-iron permease (ZIP), cation diffusion facilitator, and metal transport proteins (MTP) play a significant role in the transportation of heavy metals to vacuoles [90]. ZIP transporter proteins are involved in the uptake of Zn(II) and Fe(II) [222]. The ZIP subfamily is represented by the *Arabidopsis ZIP1*, *ZIP2,* and *ZIP3* genes and complement yeast transport mutants that show Zn (II) deficiency. In addition, during the deficiency of zinc, *ZIP1* and *ZIP3* are root genes playing an important role in zinc uptake from soil [223]. ZIP proteins passage toxic metals and nutrients as Zn(II) transport activity is repressed by Mn(II), Co(II), Cd(II), and/or Cu(II) and shows the efficiency for the transport of heavy metals. The expression of the inositol transporter (*ITR1*) gene of *Arabidopsis* increases in roots and is, therefore, used for normal iron utilization. Cd (II) and Zn (II) are efficiently transported by the ITRI protein [224]. The cation diffusion facilitator containing a protein family regulates the cation efflux far away from the cytoplasmic compartment either across the cell or into cellular compartments, such as vacuoles [225]. The cobalt (COT1) and zinc (ZRC1) proteins from *Saccaromyces cerevisiae* confer Co and Zn/Cd tolerance in plants. The inadequate information on the activation of the transcription factor functioning of metal-specific data elements indicate that plants need a range of mechanisms to activate genes so as to decrease the stress caused by the HM.

## 9. Exertion of Aquatic Macrophytes in Phytoremediation

The phytoremediation of a plant-based green technology proficiently allows plants to assemble, perfuse, and centralize contaminants. As reviewed by Hutchinson (1975), phytoremediation encompasses bio-sorption and bioaccumulation to precipitate toxins from the aquatic environment [226]. A diverse group of photosynthetic organisms in an aquatic environment can be utilized as a tool in the environmental assessment such as in situ water quality valuation due to their ability to translocate pollutants [227]. Therefore, contaminant biomonitoring in aquatic systems is an essential exertion substantially contributed to by the aquatic macrophytes [228]. The mitigation of contaminants by macrophytes is convoyed by their hasty growth and great biomass production and they act as natural filters to transport pollutants by water. These macrophytes have been universally adapted to clean polluted waters in the last few decades [229,230]. Aquatic macrophytes are most appropriate for wastewater treatment and HM accumulation in comparison to terrestrial plants. For research, particularly into the treatment of industrial and household water, these are considered to be appropriate for remediation purposes [231,232]. Their high growth ability and reproduction makes macrophytes powerful candidates for phytoremediation [233]. 

Several aquatic plants have been explored for the abatement of contaminated water with pollutants (Cu (II), Cd (II), and Hg (II)) [234,235,236].

### 9.1. Eichhornia crassipis (Water hyacinth)

Water hyacinth, due to its various capabilities, such as its fast growth, high pollution tolerance, and high absorption capacity, is frequently employed in contaminant remediation. The elimination capacity for arsenic is far more than any other macrophytes because of its great biomass content, and it thrives in all stable habitats [237]. The arsenic removal capacity of water hyacinth has been investigated by Alvarado et al. (2008), who reported that, under laboratory conditions, water hyacinth was successful in decontaminating the site with an elimination recovery of 18%. While comparing the removal efficacy rates in the tropical opencast coalmine effluent of *E. crassipes*, *Lemna minor*, and *Spirodela polyrhiza,* it has been observed that *E. crassipes* had the maximum removal efficiency (80%) in comparison to other macrophytes [237]. A recent investigation testified that *E. crassipes* accrued the maximum concentration of Pb in its tissues in comparison to its species [238]. Similarly, *E. crassipes* has been employed for the elimination of phosphate, total soluble solids (TSS), and ammonical nitrogen (NH_3_-N) [239].

Although water hyacinth is considered to be one of the most problematic plants, as reported by numerous investigations, owing to its rapid and uncontrolled growth in aquatic systems, its ability to absorb nutrients in sufficient quantities has provided new insights into its role in phytoremediation [240]. In urban and industrial areas with a high load of pollution, it can emerge as a potential pollution remediating plant, particularly in wastewater treatment. Considering the future aspect of phytoremediation, the exertion of invasive plants can assist in the sustainable management of pollution remediation of HM-contaminated sites [241].

### 9.2. Azolla caroliniana (Mosquito fern)

It has been stated that *Azolla* has a great capability to amass noxious elements (mercury, cadmium, chromium, copper, nickel, and zinc) due to its strong competence to absorb toxic heavy metals. Investigations have revealed that *Azolla* can remove pollutants from wastewater [232]. Different *Azolla* species (*A. filiculoides*, *A. microphylla*, and *A. pinnata*) have been employed for their metal (Cd, Cr, and Ni) decontamination potential. While *A. microphylla* showed greater removal efficiency for Cd, *A. pinnata* was efficient in Cr and Ni removal [242]. In other studies, it has been observed that greater Cd concentration given to *Azolla* may have a venomous effect on plant metabolic activities. Up to 0.1 mg Cd·L^−1^, plants can withstand the metal stress condition; beyond this limit an imbalance in oxidative stress and anti-oxidative enzyme production leads to decreased growth and disruptive physiological activities in *Azolla* [243].

### 9.3. Pistia stratiotes (Water lettuce)

Water lettuce has been verified as an effective plant for metal decontamination, metal depollution, and urban sewage treatment [244,245]. Due to its all-embracing root system, the roots are able to take enough metals with high removal efficiency. *Pistia stratiotes* are found to be an adequately low-cost alternative for the elimination of dissolved HMs, such as Pb and Cd of industrial effluents [246].

### 9.4. Lemnoideae (Duckweeds)

Duckweeds are profoundly present in ponds, lakes, and wetlands. Duckweed species are utilized in water eminence studies for checking HMs [247]. The plant (*Lemna* species) has a high capacity for debarring the toxic metals from water. The plant’s efficiency increases drastically at the optimum pH, which is approximately between 6 and 9, and translocates approximately 90% of soluble lead from water. However, its growth is inhibited by the increased levels of nitrate and ammonia [27]. Studies have estimated that among four metals, Cu, Cd, Pb, and Ni, accumulation and uptake of lead in the dry biomass of *L. minor* is significantly high [229]. Excellent metal efficiency was shown by plant and percentage removal was greater than 80% for all metals [229].

### 9.5. Ludwigia stolonifera

It is an exotic macrophyte that has prompt growth and multiplies at a significant rate because of its adsorbent biomass and is measured as a viable living species for the remediation of HMs [248]. As per the study [249], the plant proved to be a potential hyperaccumulator through diverse variables, untangled mechanisms of metal uptake, translocation, and transformation.

### 9.6. Salvinia (Butterfly fern)

The extensive diversity, prompt multiplication, and close linkage with other water macrophytes, including *Azolla* and *Lemna,* makes it a known choice for phytoremediation [250]. As per the reported literature, it has been stated that it poses excellent removal efficiency, particularly when exposed to glycosylate concentration [251]. *Salvinia* has also been employed for wastewater treatment [252].

### 9.7. Hydrilla verticillate (Hydrilla)

*Hydrilla verticillata* (hydrilla) is an aquatic macrophyte that forms a thick layer in the whole water body. The plant has the adeptness and potential to remove the contaminants. It has been reported that the shoots of *Hydrilla verticillata* have more ability in the translocation of toxic metal uptake instead of the roots [27]. When exposed to the high concentration of lead solution for 1 week, *Hydrilla* showed 98% uptake of lead [27]. *H. verticillata* has also shown significant potential for HM decontamination.

### 9.8. Schoenoplectus californicus (Giant bulrush)

*Schoenoplectus californicus*, also known as giant bulrush, is diverse in nature. The plant is highly permissive to high metal concentration in streams, lakes, and ponds [253]. As per the investigation conducted by the researchers, it has been estimated that shoots and roots of viable *S. californicus* sorbed 0.88% and 5.88%, respectively, in wetland treatment systems receiving copper-contaminated water [254]. Similarly, it has been demonstrated that bulrush roots accumulate the highest concentrations of pollutants, mainly dichlorodiphenyltrichloroethane (DDT) and chlordane (30.2–45.7 ng g^−1^ dry weight), and are considered suitable for the treatment of organochlorine compounds [255]. The phytoremediation prospective and HM uptake by macrophytes is shown in Table 3 and Table 4, respectively.

Even though using aquatic macrophytes for phytoremediation has provided new pathways and insights into the remediation of HMs, there are certain flaws and disadvantages associated with such a technique that need to be addressed before its application in the field. The technique of phytoremediation utilizing macrophytes for HM removal is considered to be time-consuming and can cause HM bioaccumulation in food chains that can have deleterious impacts upon the livestock as well as human health. There should be restricted access to the site. Plant species such as *Amaranthus spinosus, Alternanthera philoxeroides,* and A. *sessiles* growing on sewage sludge has been used for metal accumulation. Transfer factor and metal content in such species indicates their ability to bioconcentrate in their tissues; thus, it is possible to restore the biosolid and sewage sludge contaminated sites using these species, while exercising caution on human consumption. Similarly, *A. philoxeroides,* another edible plant used as a dietary supplement, has been used for the removal of lead and mercury from polluted waters. However, there is need to monitor the metal transfer through the food chain [189]

For the eco-rehabilitation of polluted sites, phytoremediation is emerging as a novel technique of immense potential. However, this demands a plethora of scientific research for enhancing our understanding and knowledge for the efficient remediation of HMs. The progression of omic techniques can assist in defining new metabolites and traits implicated in HM stabilization by hyperaccumulator plants which require novel strategies for its progress. Although genetic engineering has helped in HM detoxification, no perfect model of the whole data genome has yet been certified. This requires further exploration. The manipulation of microbial niches by the halogenome of the microorganisms of plants can be used to enhance resistance to HM contamination [301]. Nano remediation can be another technique of notable promise that can be employed for HM removal [302,303]. Nanoparticles derived from plants, fungi, and bacteria play an important role in remediating environmental toxic wastes [304]. The nanoparticles prove to be effective agents in cleaning up the contaminated environment as they can penetrate regions of contamination that other types of microparticles do not possess the ability to reach. These particles have higher reactivity to the contaminants in comparison to the other types of microsized particles being used for the clearing of contaminants [305]. However, there is a need to have further elucidation of the relationship between nanoparticles and molecular approaches of phytoremediation before expanding such a prospect for HM remediation [305]. Finally, the success of phytoremediation will heavily rely on the contribution and coordination of farmers, local communities, researchers, and industrial and environmental authorities. This can be achieved by imparting education programs for ensuring the extended sustainability of this green remediation technology.

## 10. Conclusions and Future Perspectives

Phytoremediation technology as a process appears to be a less disruptive, more economical, and eco-friendlier clean-up technology. Furthermore, phytoremediation needs minimal involvement of specialists, and the process can be applied for an extended time period. With the development of genetics, the accumulation and tolerance capacity of plants involved in phytoremediation can be enhanced considerably. At the molecular level, transgenic methods can be applied to augment the remediation potential of different plant species. Gene manipulation, alteration, and deletion by genetic engineering techniques have been successfully utilized to produce genetically engineered species that have shown considerably high tolerance and metal uptake capacity. The identification of quantitative trait loci and candidate genes with high biomass yield characteristics, and the subsequent development of transgenic plants with enhanced remediation potential, will encourage further research in the phytoremediation of HM-contaminated environments. It will provide new and innovative research tools for getting better results. In-depth research is warranted to discover which plant has high resistance to find its suitability for specific environmental conditions. In situ toxicity evaluation could be beneficial for the initial identification of such species. Keeping in mind the financial aspects and potential benefits, the phytoremediation technique epitomizes an effective and viable option to obtain benefits in both monetary and environmental terms in comparison to the physicochemical methods. More comprehensive investigations into the potentialities and boundaries regarding phytoremediation can enhance the practice of this technique for soil remediation in the near future.

## Figures and Tables

**Figure 1 plants-11-01255-f001:**
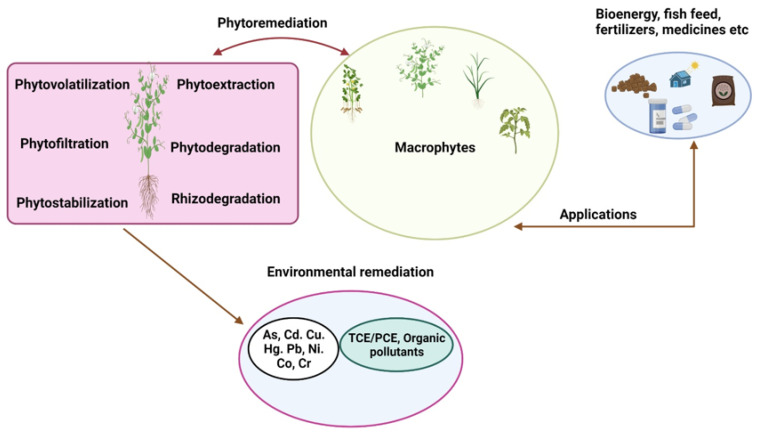
Perspectives of phytoremediation using macrophytes for the removal of heavy metals and other pollutants.

**Figure 2 plants-11-01255-f002:**
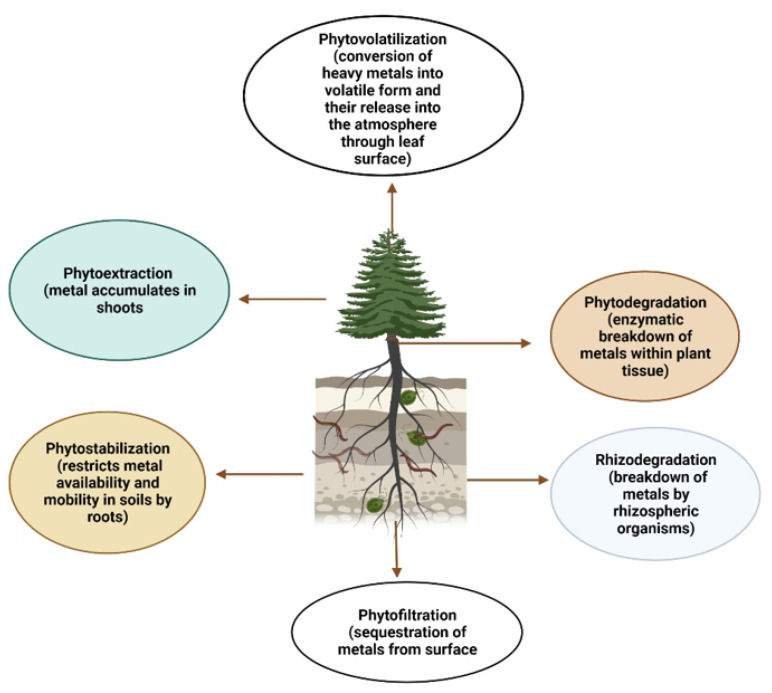
Techniques of phytoremediation and the destinies of pollutants.

**Figure 3 plants-11-01255-f003:**
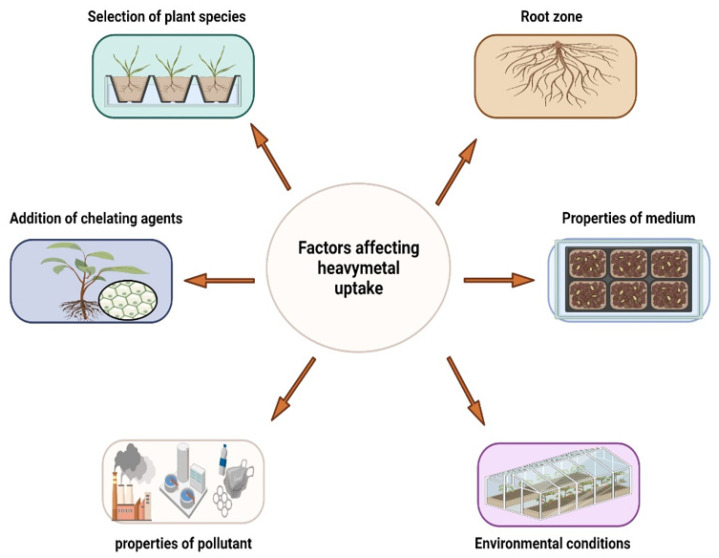
Factors affecting heavy metal uptake.

**Table 1 plants-11-01255-t001:** Application of hyperaccumulators for removal of heavy metals from contaminated soils by phytoremediation.

Hyperaccumulator	Heavy Metal	Reference
*Arabidopsis halleri*	Zn	[17,18]
*Achillea millefolium*	Hg	[19,20]
*Alyssum murale*	Ni	[21,22]
*Azolla pinnata*	Cd	[23,24]
*Thalaspi caerulescene*	Zn	[25]
*Brassica juncea* L.	Cu, Zn, Pb	[26,27]
*Brassica napus* L.	Cu, Zn, Pb	[28,29]
*Brassica oleracea, Raphanus sativus*	Zn, Cd, Ni, Cu	[30]
*Brassica nigra*	Pb	[31,32]
*Betula occidentalis*	Pb	[31,33]
*Cardaminopsis halleri*	Zn, Pb, Cd, Cu	[34]
*Cannabis sativa* L.	Cd	[35,36]
*Cicer aeritinum* L.	Cd, Pb, Cr, Cu	[37]
*Cucumis sativus* L.	Pb	[38,39]
*Eichhornia crassipes* L.	Cr, Zn	[40,41]
*Eleocharis acicularis*	As	[42,43]
*Euphorbia cheiradenia*	Pb, Zn, Cu, Ni	[44,45]
*Haumaniastrum katangense*	Cu	[45,46]
*Helianthus annuus*	Pb, Cd	[47]
*Jaltropa curcas* L.	Cu, Mn, Cr, As, Zn, Hg	[48,49]
*Lantana camara* L.	Pb	[50,51]
*Lavadula vera* L.	Pb	[52]
*Lens culunaris Medic.*	Pb	[53]
*Lepidium sativum* L.	As, Cd, Pb	[54,55]
*Lactuca sativa* L.	Cu, Mn, Zn, Ni, Cd,	[37]
*Marrubium vulgare*	Hg	[56,57]
*Miscanthus x giganteus*	Cu, Ni, Pb, Zn	[58]
*Medicago sativa*	Pb	[31,59]
*Noccaea Caerulescens*	Pb	[60]
*Oryza sativa* L.	Cu, Cd	[61,62]
*Minuartia verna, Agrostis tenius*	Pb	[63,64]
*Pelargonium*	Pb	[65,66]
*Pisum sativum* L.	Pb, Cu, Zn, Ni, As, Cr	[67]
*Potentila griffithii*	Zn	[68,69]
*Pteris vittata*	Hg	[19,70]
*Rapanus sativus* L.	Cd, Fe, Pb, Cu	[54,71]
*Salvia sclarea* L.	Pb, Cd, Zn	[69,72]
*Spinacia oleracea* L.	Cu, Ni, Zn, Pb, Cr	[73,74]
*Sorghum bicolor* L.	Cd, Cu, Zn	[72,75]
*Sorghum halepense* L.	Pb	[76,77]
*Trifolium alexandrinum*	Zn, Pb, Cu, Cd	[78,79]
*Tagetes minuta*	As, Pb	[76,80]
*Thlaspi caerulescens*	Cd	[31,81]
*Viola principis*	Pb	[82]

Pb (lead); Cr (chromium); Zn (zinc); As (arsenic); Cu (copper); Cd (cadmium); Fe (iron); Hg (mercury); Co (cobalt); Ni (nickel).

**Table 2 plants-11-01255-t002:** Exertion of soil algae for heavy metal decontamination by phytoremediation.

Alga	Heavy Metal	Reference
*Ascophyllum nodosum*	Ni, Pb	[119,120]
*Cladophora fascicularis*	Pb (II)	[121,122]
*Cladophora glomerata*	Zn, Cu	[123,124]
*Cladophora glomerata, Oedogonium rivulare*	Cu, Pb, Cd, Co	[125,126]
*Cymodocea nodosa*	Cu, Zn	[127,128]
*Fucus vesiculosis, Laminaria japonica*	Zn	[129,130]
*Oscillatoria quadripunctulata,*	Cu, Pb	[30]
*Sargassum filipendula*	Cu	[131,132]
*Sargassum natans*	Pb	[119,133]
*Spirogyra hyaline*	Cd, Hg, Pb, As	[134,135]

Pb (lead); Cr (chromium); Zn (zinc); As (arsenic); Cu (copper); Cd (cadmium); Fe (iron); Hg (mercury); Co (cobalt); Ni (nickel).

**Table 3 plants-11-01255-t003:** Heavy metal uptake by macrophytes testified in the literature.

Common Name	Scientific Name	Trace Elements	References
Duckweed	*Lemna gibba* L.	As, U, Zn	[256,257]
Lesser duckweed	*Lemna minor* L.	As, Zn, Cu, Hg	[258,259]
Water hyacinth	*Eichornia crassipes*	As, Fe, Cu, Zn, Pb, Cd, Cr, Ni, Hg	[257,259,260]
Common reed	*Phragmites australis*	Cr, Cu, Ni, Pb, S, V, Cd,	[260,261]
Water spinach	*Ipomoea aquatic*	As, Cd, Pb, Hg, Cu, Zn	[262,263]
Water fern	*Azolla filiculoides, azolla pinnata*	As, Hg, Cd	[264,265]
Elephant ear	*Colocasia esculenta*	Cd, Pb, Cu, Zn	[55,266]
Water lily	*Nymphaea violacea, Nymphaea aurora*	Cd, Pb, Cu, Zn	[23,267,268]
Water pepper	*Polygonum hydropiper*	As	[266,267]
Marshwort	*Nymphoides germinate*	Cd, Cu, Pb, Zn	[264,268]
Lesser bulrush	*Typha latifolia*	Cd, Pb, Cr, Ni, Zn, Cu	[269,270]
Brazillian waterweed	*Veronica aquatic*	As, Cr	[271,272]
Tape grass/eel grass	*Vallisneria spiralis*	Hg	[273,274]
Alligator weed	*Althernanthera philoxeroides*	As, Pb	[271,275]
Reed canary grass	*Phalaris arundinacea* L.	Pb, Zn, Cu, Cd	[276,277]
Water lettuce	*Pistia stratiotes*	As, Cr, Pb, Ag, Cd, Cu, Hg, Ni, Zn	[278,279]
Willow moss	*Fontinalis antipyretica*	Cu, Zn	[280,281]
Needle spikerush	*Eleocharis acicularis*	As, Ag, Pb, Cu, Cd, Zn, Ni, Mg	[282,283]
Rigid hornwort	*Ceretophyllum demersum*	As, Pb, Zn, Cu	[284,285]
Watercresses	*Nasturtium officinale*	Cu, Zn, Ni	[78,286]

Pb (lead); Cr (chromium); Zn (zinc); As (arsenic); Cu (copper); Cd (cadmium); Fe (iron); Hg (mercury); Co (cobalt); Ni (nickel); U (uranium); S (sulfur); Ti (titanium).

**Table 4 plants-11-01255-t004:** Macrophytes recognized for their phytoremediation prospective.

Plants	Heavy Metals	Accumulation (Dry Weight Basis)	Reference
*Eichhornia crassipes*	Hg	119ng Hg g^−1^	[287]
	Cd	3992 µg Cd g^−1^	[237]
	Cu	314 µg Cu g^−1^	[288]
	Cr	2.31 mg Cr g^−1^	[289]
	Cd	1.98 mg Cd g^−1^	[289]
	Ni	1.68 mg Ni g^−1^	[289]
*Elodea densa*	Hg	177 µg Hg g^−1^	[287]
*Lemna gibba*	Ur	897 µg Ur g^−1^	[290]
	As	1022 µg As g^−1^	[290]
*Lemna minor*	Zn	4.23–25.81 mg Zn g^−1^	[291]
	Ti	221 µg Ti g^−1^	[292]
	Cu	400 µg Cu g^−1^	[293]
	Pb	8.62 mg Pb g^−1^	[294]
*Pistia stratiotes*	Hg	83 µg Hg g^−1^	[295]
	Cr	2.50 mg Cr g^−1^	[289]
	Cd	2.13 mg Cd g^−1^	[289]
	Ni	1.95 mg Ni g^−1^	[289]
*Salvinia natans*	Cr	7.40 mg Cr g^−^^1^	[296]
*Ceratophyllum demersum*	As	525 µg As g^−^^1^	[237]
	Cd	1293 µg Cd g^−1^	[237]
	Zn	57 µg Zn g^−1^	[297]
*Potamogeton pusillus*	Cu	162 µg Cu g^−1^	[298]
*Vallisneria spiralis*	Cr	2.85 mg Cr g^−1^	[289]
	Cd	2.62 mg Cd g^−1^	[289]
	Ni	2.14 mg Ni g^−1^	[289]
	Hg	158 µg Hg g^−1^	[232]
*Myriphyllum triphyllum*	Cd	17 µg Cd g^−1^	[299]
*Sagittaria montevidensis*	Hg	62 mg Hg g^−1^	[287]
*Wolffia globose*	As	1000 µg As g^−1^	[300]
*Spirodela polyrhiza*	As	7.65 n mol As g^−1^	[282]
*Mentha* sp.	Fe	378 µg Fe g^−1^	[242]

Pb (lead); Cr (chromium); Zn (zinc); As (arsenic); Cu (copper); Cd (cadmium); Fe (iron); Hg (mercury); Co (cobalt); Ni (nickel); U (uranium); Ti (titanium).

## Data Availability

Data sharing is not applicable to this article as no new data were created or analyzed in this study.

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
