# Peer review of "Phytoremediation of Heavy Metals: An Indispensable Contrivance in Green Remediation Technology"

_plants, 2022, doi:10.3390/plants11091255_

Round 1
Reviewer 1 Report
The paper presents a good synthesis of the phytoremediation process.
L: 105: I read with interest the phytoremediation technique in situ where the advantages and disadvantages are presented and then I expected to find the same thing about phytoremediation ex situ.
L 385_445. Fro Factors affecting heavy metal uptake, There are only one or two bibliographic sources. Given the importance of these factors in depollution yields, it is considered that it is not enough to establish their range and value. For example: A. M. Chirila Babau, V. Micle, G. E. Damian, I. M. Sur. Preliminary investigations on the potential of Robinia pseudoacacia L. (Leguminosae) in phytoremediation of waste dumps. Journal of Environmental Protection and Ecology 21, No 1, 46-55 (2020), Soil Pollution
It has been specified that it is not recommended to study plants that can be consumed by humans and animals. Some examples should be specified, given that they exist in the literature.
Author Response
the suggestions/ recommendations by the worthy reviewer has been incorporated.
kindly find the detailed response t the queries in the attachment

Reviewer 2 Report
The green remediation of environmental heavy metal pollution is a very practical research content. The topic of this paper is very in line with the academic frontier. The author consulted a large number of relevant literature and made a good summary. I agree to publish this article after modification.
1, This paper mainly summarizes the remediation of heavy metal pollution, and "heavy metal" must be added to the title of the article.
2, "As heavy metals exist in our environment as persistent pollutants, they require complete removal." This sentence is wrong. Heavy metals cannot be completely removed as single elements. And It is unlikely to be completely removed.
3, The quality and layout of the Figures are too poor. There is still a huge space for beautification of all three Figures.
4, The latest literature on the green technology of repairing heavy metals with many nano materials is missing and insufficiently discussed.
Application of Nanoparticles Alleviates Heavy Metals Stress and Promotes Plant Growth: An Overview.NANOMATERIALS, 11(1):26
Magnetic (Fe3O4) Nanoparticles Reduce Heavy Metals Uptake and Mitigate Their Toxicity in Wheat Seedling. SUSTAINABILITY, 9(5):790
Author Response
The detailed response to the queries/ recommendations suggested by the worthy reviewer is given in the attached file

Round 2
Reviewer 2 Report
Authors have addressed all my comments.
Author Response
Thankyou very much for your worthy suggestions and recommendations